# Pulmonary Complications after COVID-19

**DOI:** 10.3390/life12030357

**Published:** 2022-02-28

**Authors:** Petr Jakubec, Kateřina Fišerová, Samuel Genzor, Milan Kolář

**Affiliations:** 1Department of Pulmonary Diseases and Tuberculosis, Faculty of Medicine and Dentistry, Palacký University Olomouc, Hněvotínská 3, 77900 Olomouc, Czech Republic; petr.jakubec@fnol.cz (P.J.); samuel.genzor@fnol.cz (S.G.); 2Department of Pulmonary Diseases and Tuberculosis, University Hospital Olomouc, I. P. Pavlova 185/6, 77900 Olomouc, Czech Republic; 3Department of Microbiology, Faculty of Medicine and Dentistry, Palacký University Olomouc, Hněvotínská 3, 77900 Olomouc, Czech Republic; milan.kolar@fnol.cz

**Keywords:** long COVID-19, pulmonary complications, bacterial respiratory infections

## Abstract

Coronavirus disease 2019 (COVID-19) is a threat to patients not only because of its acute course, but also because of various complications occurring in the following period, that is, more than 28 days after the onset of acute infection. The present study identified a total of 121 patients hospitalized 29 or more days after the first positive result of a PCR test for SARS-CoV-2, of whom 98 patients were included in the study. Patients were divided into two groups by the time interval between the positive COVID-19 test result and hospitalization date. The time intervals were week 5–11 in an ongoing-COVID group (57.1% of patients) and 12 or more weeks in a post-COVID-group (42.9%). The most frequent reason for hospitalization was respiratory tract infection (58.2%). Pneumonia accounted for 77.2% of these cases. Other reasons for hospitalization were interstitial lung disease (22.4%), pulmonary embolism (8.2%), and sarcoidosis (6.1%). The study group was further divided according to the causes of hospitalization into subgroups with infections and other causes. In the group with infectious diseases, there was a shorter time period between PCR positivity and hospitalization and there were significantly more frequent non-respiratory complications. In the entire sample, the in-hospital mortality was 5.1%.

## 1. Introduction

Acute infection with coronavirus disease 2019 (COVID-19) persists for up to four weeks after the initial symptoms develop. The period after acute infection, referred to as long COVID-19 by the NICE, is divided into two phases, ongoing (post-acute) COVID-19 present from four up to 12 weeks and post-COVID-19 present for more than 12 weeks [1]. Post-COVID-19 complications are defined as (a) persistent, worsening, re-emerging or new acute infection symptoms, (b) deterioration of the quality of life or functional status compared with the pre-COVID-19 period, (c) the presence of persistent or progressive pulmonary pathology on radiologic imaging or abnormal lung function test results with other possible causes being ruled out [2]. Recently, the term persistent post-COVID syndrome has been used in the literature, defined as an entity involving physical, medical and cognitive sequelae of COVID-19 infection, including persistent immunosuppression and pulmonary, cardiac and vascular fibrosis. The syndrome leads to increased vulnerability to secondary infections and organ dysfunction even after a seeming recovery from acute COVID-19 infection [2]. During COVID-19, systemic inflammatory response syndrome develops excessively, marked by elevated pro-inflammatory cytokine levels. To achieve balance, the body responds by developing compensatory anti-inflammatory response syndrome. However, if the response is inadequately strong, protracted immunosuppression follows, referred to as persistent inflammation, immunosuppression and catabolism syndrome [3]. Such patients are at increased risk for bacterial and fungal infections, and also likely to develop pulmonary fibrosis [4,5,6,7].

Post-COVID-19 complications occur in 10–35% of individuals treated at home and as many as 80% of inpatients [8,9,10,11,12,13]. According to numerous studies, the symptoms may persist for more than four weeks after the infection in 33–87% of patients and 12 or more weeks after the infection in 25–87% of patients, regardless of the disease severity [14,15,16,17,18,19,20,21,22,23]. These complications may be respiratory, cardiovascular, nasopharyngeal, neuropsychiatric, gastrointestinal, musculoskeletal, endocrinal, dermatological and, rarely, ophthalmic.

The present study aimed to assess serious pulmonary complications requiring hospital (re)admission from home care in patients with long COVID-19, that is, at least 29 days from the first positive result of a PCR test for SARS-CoV-2. We describe the demographic characteristics of the study group, time interval between the positive COVID-19 test result and hospital admission date, admission reasons, laboratory parameters, complications, and comorbidities. Statistical analyses were performed, including comparisons between the post-COVID and ongoing-COVID groups as well as between the groups with infective and other causes of admission.

## 2. Materials and Methods

Between 1 November 2020 and 31 July 2021, a total of 121 patients admitted to the Department of Pulmonary Diseases and Tuberculosis, University Hospital Olomouc, Czech Republic met the following criteria: (1) COVID-19 infection, (2) 29 or more days from a positive test and (3) admission from home care. Of those, 98 were included in the study. The reasons for not including the remaining 23 patients were non-respiratory complications in eight cases (three of gastrointestinal tract infections, two of cardiac failure, and one each of urinary tract infection, myasthenia gravis and fatigue syndrome). Also excluded were 15 patients with respiratory diseases apparently not associated with COVID-19 infection (five with bronchogenic carcinoma, five with tracheal stenosis, three with pulmonary fibrosis, and one each with pulmonary emphysema and postoperative pulmonary complications).

Patients were divided into two groups by the time interval between the positive COVID-19 test result and hospital admission date. The time intervals were week 5–11 (28–77 days) in the ongoing-COVID group and 12 or more weeks (>77 days) in the post-COVID-group.

The primary diagnoses for which patients were admitted to the department were made based on their history, clinical examination, laboratory test and imaging results. Infections were diagnosed by the presence of clinical signs of lower respiratory infection, elevated laboratory markers of inflammation (white blood cell count, C-reactive protein) and, in case of pneumonia, new or progressive infiltrates on chest radiographs or lung CT scans. Non-infective chronic obstructive pulmonary disease (COPD) exacerbations were confirmed by auscultation (prolonged expiratory time, expiratory sounds, or wheezing and crackles), low levels of inflammatory markers and no infiltrates on chest radiographs or lung CT scans. Interstitial lung disease (ILD) was diagnosed by chest radiographs or lung CT scans (ground-glass opacity, reticular patterns, crazy paving, condensations typical of organizing pneumonia) and ruling out other causes (e.g., pneumonia, pulmonary embolism). Pulmonary embolism was confirmed by a CT pulmonary angiogram or ventilation/perfusion lung scan. The diagnosis of sarcoidosis was based upon clinical suspicion, laboratory findings, typical findings on chest radiographs or lung CT scans (mediastinal lymphadenopathy, nodular lung disease typically in the peribronchial or perilymphatic area), bronchoalveolar lavage fluid analysis results, and always confirmed by typical histological findings of non-caseating epithelioid granuloma.

Furthermore, the following laboratory parameters were analyzed: C-reactive protein (CRP), procalcitonin (PCT), white blood cells (WBC), D-dimers, urea, creatinine, troponin, N-terminal prohormone of brain-natriuretic peptide (NT-proBNP), total protein (TP), albumin, pH, partial pressure of oxygen (pO_2_) and partial pressure of carbon dioxide (pCO_2_). Other evaluated complications were pneumothorax, pleural effusion, bronchiectasis, ILD, pulmonary embolism and non-respiratory complications: cardiovascular complications generally, arrhythmias, heart failure, renal failure, myopathy and *Clostridioides difficile* infection. The presence of comorbidities was assessed: arterial hypertension, cardiovascular disease, respiratory diseases (COPD, bronchial asthma, ILD, cystic fibrosis).

Clinical samples from the respiratory tract (nasopharyngeal swabs, sputum, airway secretions) were regularly collected (on admission and then at least once a week) from all patients. Bacterial pathogens were identified using standard microbiology techniques with a MALDI-TOF MS system (Biotyper Microflex, Bruker Daltonics). For each patient, a single strain of each species, isolated as the first one from a particular clinical sample, was included in the study. Susceptibility to antibiotics was determined with a standard microdilution method in accordance with the EUCAST criteria [24]. To ensure quality control, the following reference bacterial strains were used, *Escherichia coli* ATCC 25,922, *Pseudomonas aeruginosa* ATCC 27,853, *Staphylococcus aureus* ATCC 29,213 and *Enterococcus faecalis* ATCC 29,212.

The IBM SPSS Statistics Software for Windows, Version 23.0. Armonk, NY, USA: IBM Corp was used for statistical evaluation. All tests were performed at a level of significance of 0.05. The Shapiro–Wilk test was used for the evaluation of normality of the distribution, Mann–Whitney *U*-test was used for comparing quantitative parameters, Fisher’s exact test was used to compare percentage proportions of parameters between the study subgroups.

## 3. Results

The sample of 98 patients aged 21–93 years comprised 65 males (66.3%) and 33 females (33.7%). The median time between positive results of PCR tests for SARS-CoV-2 and admission to the department was 60.0 days (29–252 days). There were 56 patients (57.1%) in the ongoing-COVID group and 42 patients (42.9%) in the post-COVID-group. In the ongoing-COVID group, the median time from positive PCR tests to admission was 103 days (78–252 days). The median age of patients in the ongoing-COVID group (69.0 years) was slightly higher compared to that in the post-COVID group (67.0 years), but the difference was not statistically significant (*p* = 0.897). In the post-COVID group, there was a non-significantly higher proportion of males (71.4% compared with 62.5% in the ongoing-COVID group, *p* = 0.394).

The participants were subdivided based on the main reason for hospital admission (Table 1).

The most frequent reason for hospital admission was respiratory infection in 58.2% of patients (67.9% in the ongoing-COVID group and 45.2% in the post-COVID group), followed by ILD in 22.4% (14.3% and 33.3%, respectively). Other causes were pulmonary embolism (8.2%), mainly in the post-COVID group, and non-infective COPD exacerbations, equally present in both subgroups. Among infective causes, pneumonia was most common, accounting for 77.2% of all infective process cases (71.1% in the ongoing-COVID group and as many as 89.5% of cases in the post-COVID group. Other infective causes were less frequent (Table 2).

No significant differences in age (68.0 vs. 68.0; *p* = 0.798) or sex ratio (70.2% of males vs. 61.0%; *p* = 0.390) were found between the groups with infective and non-infective causes of admission. A possible (but above the significance level) trend was found in the parameter of the time from the positive result of PCR test of SARS-CoV-2 to admission, where median time in infectious causes was shorter than in non-infectious causes (50.0 days vs. 78.0 days in non-infectious causes, *p* = 0.053).

When including all pulmonary complications, that is, both those requiring hospital admission and other accompanying respiratory complications, ILD was found in 57.1% of patients (56/98), pulmonary embolism in 13.3% (13/98), pleural effusion in 12.2% (12/98), bronchiectasis in 10.2% (10/98) and pneumothorax in 3.1% (3/98).

Comparison of types of pulmonary damage found no significant differences, the only exception being bronchiectasis which was found to be more common in the post-COVID group (4% vs. 19%; *p* = 0.017) (Table 3).

Non-infective pulmonary complications were significantly more frequent in case of ILD (47.3% in patients with infections vs. 73.2% in those with non-infective causes of admission; *p* = 0.013). Significantly more frequent were also pleural effusions (21.1% in the group with infective causes of admission vs. 0% in the group with other causes, *p* = 0.001) (Table 4).

Statistically significant differences in laboratory parameters were found between the groups with infective and other causes of admission, namely in the levels of TP (63.8 g/L vs. 68 g/L; *p* = 0.034), albumin (35.9 g/L vs. 42.1 g/L; *p* < 0.0001), pO_2_ (7.8 kPa vs. 8.8 kPa; *p* = 0.018). No other significant differences were found in the other laboratory parameters (Table 5).

In the group with infective causes of admission, significant differences were found between the ongoing-COVID and post-COVID subgroups, namely in D-dimers (2776.7 µg/L vs. 1101.8 µg/L; *p* = 0.003), NT-proBNP (1813.0 ng/L vs. 1253.7 ng/L; *p* = 0.039) and albumin (34.3 g/L vs. 39.1 g/L; *p* = 0.003). No other significant differences were found in the other laboratory parameters (Table 6).

Apart from respiratory complications, cardiac and renal complications were most common (11 patients, 11.2%) (Table 7 and Table 8).

There were no significant differences in the mean number of non-respiratory complication between the ongoing- and post-COVID groups (0.6 vs. 0.8; *p* = 0.984). On the other hand, the mean number of non-respiratory complications was significantly higher in the subgroup with infective causes of admission (0.7 vs. 0.2; *p* = 0.005).

Comorbidities were highly prevalent in the entire patient sample. The most frequent comorbidities were hypertension in 56 patients (57.1%), chronic respiratory diseases (COPD, bronchial asthma, ILD, cystic fibrosis) in 41 patients (41.8%), diabetes mellitus in 38 patients (38.8%) and cardiovascular diseases in 30 patients (30.6%). The proportion of patients with malignancies was also high (22.4%). (Table 9 and Table 10).

There were no significant differences in the mean number of comorbidities between the ongoing- and post-COVID groups (2.4 vs. 1.9; *p* = 0.312). Similarly, no significant differences were found between the subgroups with infective and other causes of admission (2.2 vs. 2.2; *p* = 0.696).

Oxygen therapy was required by 51 patients (52.0%) in the entire sample. In the vast majority of cases (49/51), standard oxygen therapy was administered. One patient needed high-flow nasal oxygen therapy and another one required non-invasive ventilation. In the entire sample, five patients (5.1%) died, four (7.1%) in the ongoing-COVID group and only one (2.4%) in the post-COVID group.

Regarding the etiology of bacterial infective complications, polymicrobial etiology (35.1%) slightly prevailed over monomicrobial etiology (26.3%). Etiologic agents could not be identified in 38.6% of patients. The most frequent group of detected bacterial pathogens were enterobacteria, accounting for 41.1% of all 73 etiologic agents. In this group of bacteria, 13 (43.3%) strains producing extended-spectrum beta-lactamases (ESBLs) were identified. Production of AmpC beta-lactamases, carbapenemases and metallo-beta-lactamases was not detected. The most common pathogenic enterobacterium (11.0% of all bacterial pathogens), as well as the bacterium with the highest number of strains producing ESBLs (46.2% of all ESBL-positive strains) was *Klebsiella pneumoniae*. Other most frequently isolated enterobacteria were *Enterobacter cloacae* (9.6%), *Escherichia coli* (5.5%), *Klebsiella variicola* (4.1%) and *Klebsiella oxytoca* (2.7%). However, the most frequent etiologic bacterial species was *Pseudomonas aeruginosa* detected in 12 patients (16.4%), most commonly diagnosed with pneumonia (eight cases). In this pathogen, only one (8.3%) multidrug-resistant strain was detected. The second most commonly identified bacterial pathogen was *Chlamydophila pneumoniae*, present in 10 patients (13.7%), mainly those with pneumonia (six cases). Another frequently isolated bacterial pathogen was *Staphylococcus aureus*; it was also confirmed in blood culture from a patient with pneumonia. Methicillin-resistant *Staphylococcus aureus* was not detected. Among enterococci (6.8% of all isolates), no vancomycin-resistant strain was found. Multidrug antibiotic resistance was shown in 19.2% of all identified bacterial pathogens. The percentages of bacterial species are shown in Figure 1.

## 4. Discussion

The characteristics of post-COVID-19 syndromes vary widely. The most common are persistent fatigue (47–87%), exertional dyspnea (10–71%), cough (7–10%), diarrhea (6%), olfactory (14%) and gustatory disorders (7%), muscle pain (25%), joint pain (20%), headache (18%), chest pain (15%), sleep disorders (18–26%) and neuropsychiatric disorders (4–33%) [19,25,26,27,28,29,30,31,32].

Cardiovascular complications such as myocarditis, pericarditis, myocardial infarction, arrhythmia and pulmonary embolism may develop even weeks after acute infection [13]. These are more common in patients with pre-existing cardiovascular disease [33]. Silent but progressive myocardial injury may result in cardiac failure or other cardiovascular diseases [34]. Lu et al. found that after three-month follow-up, many patients had micro-structural and metabolic changes in the brain correlated with persistent neurological symptoms such as memory loss, smell loss and fatigue [35].

Renal impairment with a decrease in glomerular filtration below 90 mL/min (despite having normal rates in the acute phase and no acute kidney injury) was present in 13% of patients after severe COVID-19 [26]. In another study, hemodialysis was needed in 8% of subjects eight weeks after severe COVID-19 infection [36].

A new nosological entity, COVID-19-associated nephropathy, has been described; it is a variant of focal segmental glomerulosclerosis and acute tubular injury [37]. COVID-19 infection may also cause intestinal dysmicrobism, potentially leading to the overgrowth of opportunistic infectious pathogens and development of *Clostridioides difficile* infection [38,39,40]. As for endocrine diseases, diabetes mellitus may newly develop weeks or months after acute infection [41]. Possible but rare complications are Hashimoto’s thyroiditis or Graves’ disease [42,43].

A recent study of 40,000 COVID-19 patients showed an increased risk for new-onset respiratory and cardiovascular diseases and diabetes mellitus over a mean follow-up of 140 days as compared to controls [44]. Persistent post-COVID-19 symptoms are significantly associated with age, female gender, hospital stays, disease severity, initial dyspnea, need for oxygen therapy, comorbidities, in particular hypertension, and chronic lung diseases [12,45,46,47,48].

The most common pulmonary post-COVID-19 symptoms are dyspnea and cough. According to Chopra et al., 7% of COVID-19 patients are left with persistent hypoxemia requiring oxygen therapy or continuous positive airway pressure or other ventilatory support during their sleep [20]. After acute COVID-19 infection, the most frequent CT findings are ground-glass opacity and bilateral consolidation with peripheral and diffuse distribution, traction bronchiectasis and reticular patterns [49]. The most frequent lung function impairment is reduced diffusing capacity, with obstructive and restrictive ventilatory defects being much rarer. Lung function impairment may persist for a long time, especially in pulmonary fibrosis or pulmonary bloodstream disease [50,51]. Studies assessing morphological and functional changes in the lungs have yielded rather variable data, with significant structural pathologies in 19–91% and functional disorders in 9–58% of the subjects [26,52,53,54,55,56,57].

Pulmonary complications accompanying the ongoing-COVID-19 phase are mainly secondary infections, lung function impairment, pulmonary thromboembolic disease (pulmonary embolism, stroke), pulmonary hypertension, pulmonary fibrosis, cavitary lesions and small airway disease [58,59]. In the post-COVID-19 period, lung function impairment, in particular reduced diffusing capacity, ILD including pulmonary fibrosis, bronchiectasis, tracheomalacia and small airway disease are most frequently observed.

There are also studies on the frequency and causes of hospital (re)admissions in the long COVID-19 period. However, a large meta-analysis involving 266,677 patients followed up for 10 to 365 days after hospital discharge only presented data on readmissions and post-discharge all-cause mortality after the first hospital stay for acute COVID-19 infection, but did not specify the reasons for hospital readmissions. At 30 days, 90 days and one year from discharge, the readmission rates were 9% (95% CI: 7.44, 10.50), 10% (95% CI: 8.37, 11.24) and 10% (95% CI: 8.92, 11.77), respectively, and the mortality rates were 8% (95% CI: 2.78, 12.96), 8% (95% CI: 4.73, 10.53) and 8% (95% CI, 5.30, 9.72), respectively [60].

Numerous studies have evaluated hospital readmissions after COVID-19 infection over a certain time period. Several studies found a very short median time to readmission of five to eight days [61,62,63,64]. The proportion of readmitted persons was 4.2–9.0%. The reasons for hospital readmissions were progression of lung involvement due to COVID-19 pneumonia, respiratory failure, unspecified infectious diseases, cardiovascular diseases, venous thrombosis, and gastrointestinal diseases. Other studies evaluated readmissions over a period of 60 days after confirmed infection with SARS-CoV-2; the proportion of readmission was 4.5–19.9% [65,66,67]. The most common causes of admissions were respiratory failure, progression of inflammatory changes after previous COVID-19 infection, venous thromboembolism, and various infective complications.

Clark et al. followed up 466 patients at 30 days and four months from discharge. Within 30 days, 28 patients (42%) were readmitted; another 41 persons (48%) were readmitted between 31 days and four months. The most common reasons for hospital readmissions within 30 days were exacerbations of COVID-19 infection with or without respiratory distress (29%), bacterial pneumonia (14%), pulmonary issues including asthma or COPD exacerbations, pulmonary edema, pleurisy, and airway stenosis (14%), bleeding (7%) and other infections (7%). Over the other time period, pulmonary issues (20%), acute renal dysfunction (15%), stroke or transient ischemic attack (12%), unspecified infection (12%) and cardiac failure (9.8%) were the most frequently observed [68].

Several studies monitored a period of six months after hospital discharge. Readmissions were present in 4.4–26.8% of persons, the most common causes being respiratory disorders, cardiovascular diseases, neuropsychiatric causes, and renal impairment. All infections caused 7.9–14% of readmissions [69,70,71].

However, none of the above studies provided detailed analysis of respiratory infective complications. The rates of infections causing hospital readmissions are rather low, if stated at all, in those studies. This is in sharp contrast to the present study, in which infections were the most frequent reason for hospital readmissions (58% of cases), with pneumonia accounting for as many as 77% of them. The only exception is the study by Parra et al. [67] reporting pneumonia as the cause of readmission in 56% of patients; however, given the rather short time interval between hospital discharge and readmission, it may be assumed that a significant proportion of those cases were exacerbations of COVID-19 infection. Even a detailed literature search failed to find a study primarily concerned with pulmonary post-COVID-19 complications and especially infections requiring hospital admissions or readmissions of patients in the long COVID-19 period. In that respect, the present study may be the first of its kind. Complicating infections are known to considerably affect the course of recovery from COVID-19 [72]. More frequent secondary infections in the long COVID-19 period may be explained by immunosuppression caused by systemic inflammatory response, similar to that following sepsis [2].

The median age in our study group was 68 years. There were no significant differences in the median age between the ongoing- and post-COVID groups or between the subgroups with infective and other causes of admission. The proportion of males was non-significantly higher in all groups. In the ongoing-COVID-19 group, the median time from PCR-confirmed SARS-CoV-2 infection was 38.5 days compared with a median of 103 days in the post-COVID group. In the group with infective causes of admission, there was a trend towards a shorter time to admission (50.2 days vs. 78 days, *p* = 0.053).

There were no significant differences in pulmonary damage types between the ongoing- and post-COVID groups, with the exception of bronchiectasis, which was more frequent in the post-COVID group. This finding is not surprising, as bronchiectasis usually develops after a longer time. Evaluation of complications showed significantly more frequent ILD in the group with other causes of admission. Also not surprisingly, pleural effusions were significantly more frequent in the group with infective causes of admission, illustrating the severity of the infections affecting the pleura as well.

In the present study, infective complications were most commonly caused by strains of *Pseudomonas aeruginosa*, *Chlamydophila pneumoniae*, *Klebsiella pneumoniae*, *Enterobacter cloacae*, *Staphylococcus aureus* and *Escherichia coli*. Other bacterial species were identified in less than 5% of cases. In 43% of enterobacterial isolates (mostly *Klebsiella pneumoniae* strains), ESBL production was detected. In only one case (8%), a multidrug-resistant strain of *Pseudomonas aeruginosa* was identified. Among Gram-positive bacteria, no strains of methicillin-resistant *Staphylococcus aureus* or vancomycin-resistant enterococci were detected. To the best of our knowledge, no study has been published that characterizes bacterial pathogens, including their resistance to antibiotics, in patients with pulmonary post-COVID-19 complications. The present study results may inspire other authors to publish similar articles.

Interstitial lung disease of varied clinical significance is a common post-COVID-19 complication. This may range from completely asymptomatic post-inflammatory changes to advanced pulmonary fibrosis. At present, no reliable data on the prevalence and severity of COVID-19-associated pulmonary fibrosis are available [73]. Regarding the pathophysiology, it is assumed that in the first phase, the main role is played by the virus itself, together with pneumonitis, a hypercoagulable state, massive pro-inflammatory response, and acute respiratory distress syndrome; sometimes, hyperoxia and ventilator-induced lung injury or patient self-induced lung injury may contribute. The second phase is dominated by immune dysregulation with compromised repair of lung tissue and its fibrosis [70]. According to some authors, the risk factors for the development of pulmonary fibrosis include older age, severe dyspnea, tachypnea, hypertension, intensive care unit stays, acute respiratory distress syndrome, invasive ventilation therapy, lymphocytopenia and high CRP levels [74,75,76]. The adverse prognostic factors are advanced age, comorbidities, smoking, obesity, genetic predisposition, pregnancy, extremely elevated inflammatory markers, lymphocytopenia and refractory hypoxemia [72,77,78,79,80]. Even though the present study found ILD in 57% of patients, all CT scans showed organizing pneumonia, ground-glass opacities, reticulations or combinations of these findings. There was not a single case of fibrotic changes.

Bronchiectasis was found on CT scans of 10–23% of patients with previous COVID-19 infection [81]. In the present study, bronchiectasis was detected in 10% of patients; not surprisingly, significantly more frequently in the post-COVID group. Pulmonary embolism may develop in both the ongoing- and post-COVID-19 periods [82,83,84]. Thromboembolic post-COVID-19 complications are not frequently reported, the incidence being 1–3% after 30–44 days from hospital discharge [12]. In the present study, however, pulmonary embolism occurred in 13% of patients, suggesting that hypercoagulable states could persist for as long as several weeks after acute infection. Interestingly, there were more than few cases of newly diagnosed sarcoidosis (6%) in the present study. All individuals with sarcoidosis were found to have new respiratory symptoms (cough, exertional dyspnea, chest oppression) over a period of 2–6 months from PCR-confirmed SARS-CoV-2 infection. In all subjects, high-resolution CT scans of the lungs revealed mediastinal lymphadenopathy, with four also having multiple micronodules in the lungs. Is it possibly because SARS-CoV-2 influences, in some unknown way, the immune system, leading to the development of sarcoidosis. To answer this question, further studies and more detailed research would be needed.

There were significant differences in the levels of laboratory parameters between the groups with infective and other causes of admission, namely in TP, albumin and pO_2_. Lower levels of TP and albumin are typical for more severe inflammation. Lower levels of oxygen may be caused by a combination of the infection itself and common ILD as well as more common comorbidities. In the group with infective causes of admission, there were significant differences in the levels of D-dimers, NT-proBNP and albumin. The explanation may be a prolonged hypercoagulable state after acute COVID-19 and gradually regressing irritation of the myocardium. However, the level of troponin was not significantly higher in the ongoing-COVID group, but it was close to the level of significance. There were no significant differences in other laboratory parameters.

The vast majority of observed non-respiratory complications were present in infective processes. The most common complications (in 11% of patients) were cardiac and renal. Among the former, cardiac failure was most frequently seen. Renal disease was clearly the most common complication in the ongoing-COVID group (14% of patients).

The presence of non-respiratory complications was not significantly different between the ongoing- and post-COVID groups. On the other hand, non-respiratory complications were significantly more frequent in the group with infective causes of admission compared to that with other causes (0.7 vs. 0.2; *p* = 0.005).

Comorbidities were rather common in the entire sample, the most frequent being arterial hypertension (57%), followed by chronic respiratory diseases (42%), diabetes mellitus (39%) and cardiovascular diseases (31%). Additionally, malignancies, both pulmonary and non-pulmonary, were frequent (22%). The absolute number of comorbidities was highest among patients admitted for infections, but the numbers of comorbidities were not significantly different between the ongoing- and post-COVID groups and between the groups with infective and other causes of admission.

In the entire sample, the in-hospital mortality was 5%. Five patients died, four (7%) in the ongoing-COVID group and only one (2%) in the post-COVID group, but there was no significant difference between the ongoing- and post-COVID groups (7.3% vs. 2.4%; *p* = 0.385). There was a trend towards higher mortality in the group with infective causes of admission, but the difference did not reach the level of statistical significance (8.9% vs. 0%, *p* = 0.071). In all five cases, the main cause was infection, namely pneumonia and bronchitis in three and two patients, respectively.

The limitation of the study is the relatively small patient sample. On the other hand, the single-center design may be considered as advantage as all subjects were evaluated by the same physicians and the laboratory and imaging methods were performed with the same equipment.

## 5. Conclusions

Complications after acute COVID-19 infection occurring in the so-called long COVID-19 period are common and rather varied. They affect numerous organs and manifest with a range of symptoms. Their course and severity are very diverse. Rather frequent and clinically most important are respiratory complications, not uncommonly requiring hospital admission. In the present study, these severe respiratory post-COVID-19 complications were most frequently caused by bacterial respiratory infections, especially pneumonia. It appears that these infective complications are more frequent compared to other respiratory complications. The most common non-infective respiratory complication is ILD, which is not always the primary cause of hospital admission, but may accompany other more serious lung disorders. The respiratory complications, especially infective respiratory diseases, were also most often accompanied by non-respiratory complications, occurred in patients with most numerous comorbidities and had the worst prognosis.

## Figures and Tables

**Figure 1 life-12-00357-f001:**
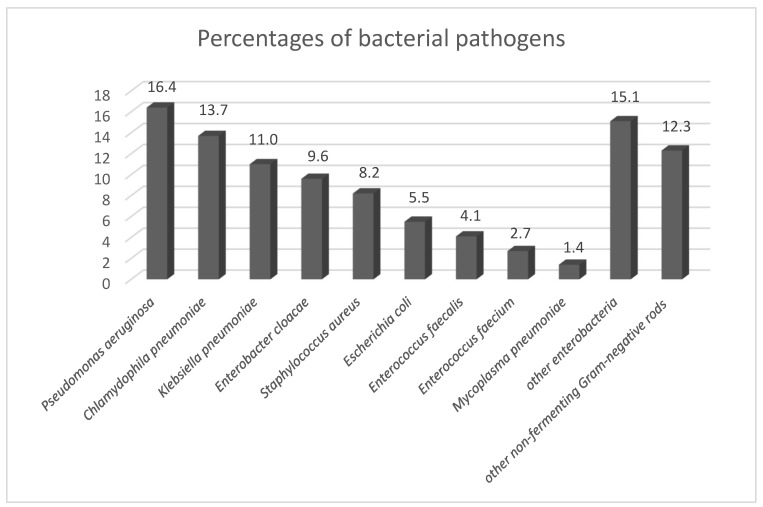
Percentages of bacterial pathogens. Legend: other enterobacteria = *Klebsiella variicola* (4.1%), *Klebsiella oxytoca* (2.7%), *Citrobacter freundii* (1.4%), *Morganella morganii* (1.4%), *Proteus mirabilis* (1.4%), *Klebsiella aerogenes* (1.4%), *Enterobacter bugandensis* (1.4%), *Serratia marcescens* (1.4%); other non-fermenting Gram-negative rods = *Achromobacter xylosoxidans* (4.1%), *Burkholderia multivorans* (2.7%), *Stenotrophomonas maltophilia* (1.4%), *Burkholderia cenocepacia* (1.4%), *Acinetobacter baumannii* (1.4%), *Acinetobacter pitii* (1.4%).

**Table 1 life-12-00357-t001:** The main diagnoses of post-COVID-19 pulmonary complications upon hospital admission.

Diagnosis on Admission	Total (*n* = 98)	Ongoing-COVID (*n* = 56)	Post-COVID (*n* = 42)
Infection	57 (58.2%)	38 (67.9%)	19 (45.2%)
COPD exacerbation, non-infective	5 (5.1%)	3 (5.4%)	2 (4.8%)
ILD	22 (22.4%)	8 (14.3%)	14 (33.3%)
Pulmonary embolism	8 (8.2%)	6 (10.7%)	2 (4.8%)
Sarcoidosis	6 (6.1%)	1 (1.7%)	5 (11.9%)

**Table 2 life-12-00357-t002:** Infective causes of hospital admissions.

Infection	Total (*n* = 57)	Ongoing-COVID (*n* = 38)	Post-COVID (*n* = 19)
Pneumonia	44 (77.2%)	27 (71.1%)	17 (89.5%)
COPD exacerbation, infective	7 (12.3%)	6 (15.8%)	1 (5.3%)
Bronchitis	6 (10.5%)	5 (13.2%)	1 (5.3%)

**Table 3 life-12-00357-t003:** Pulmonary complications in the groups of patients.

		Group				*p*
		On Going		Post COVID		
		Count	Percentage	Count	Percentage	
Sarcoidosis	yes	1	1.8%	5	11.9%	0.081
	no	55	98.2%	37	88.1%	
Pneumonia	yes	27	48.2%	17	40.5%	0.539
	no	29	51.8%	25	59.5%	
ILD	yes	30	53.6%	27	64.3%	0.309
	no	26	46.4%	15	35.7%	
Pneumothorax	yes	2	3.6%	1	2.4%	1.000
	no	54	96.4%	41	97.6%	
Pleural effusion	yes	9	16.1%	3	7.1%	0.225
	no	47	83.9%	39	92.9%	
Pulmonary embolism	yes	8	14.3%	5	11.9%	0.774
	no	48	85.7%	37	88.1%	
COPD exacerbation	yes	8	14.3%	4	9.5%	0.547
	no	48	85.7%	38	90.5%	
Bronchitis	yes	7	12.5%	1	2.4%	0.133
	no	49	87.5%	41	97.6%	
Bronchiectasis	yes	2	3.6%	8	19.0%	0.017
	no	54	96.4%	34	81.0%	

**Table 4 life-12-00357-t004:** Pulmonary complications in the groups with both infective and other causes of admission.

		Cause of Admission				*p*
		Infection		Other		
		Count	Percentage	Count	Percentage	
ILD	Yes	27	47.4%	30	73.2%	0.013
	No	30	52.6%	11	26.8%	
Pneumothorax	Yes	2	3.5%	1	2.4%	1.000
	No	55	96.5%	40	97.6%	
Pleural effusion	Yes	12	21.1%	0	0.0%	0.001
	No	45	78.9%	41	100.0%	
Pulmonary embolism	Yes	5	8.8%	8	19.5%	0.141
	No	52	91.2%	33	80.5%	
Bronchiectasis	Yes	5	8.8%	5	12.2%	0.738
	No	52	91.2%	36	87.8%	

**Table 5 life-12-00357-t005:** Laboratory parameters in the groups with infective and other causes of admission.

	Causes of Admission												*p*
	Infection						Other						
	Median	Min	Max	Mean	SD	N	Median	Min	Max	Mean	SD	N	
D-dimer	1253.0	248.0	10,375.0	2233.5	2427.7	37	1152.0	202.0	5344.0	1677.2	1650.8	17	0.407
Urea	5.6	2.4	23.9	7.4	4.5	57	6.2	3.4	20.0	7.6	3.9	41	0.306
Creatinine	82.0	6.1	239.0	89.3	40.5	57	81.5	54.0	176.0	90.5	31.0	40	0.750
Troponin	13.0	3.0	71.0	18.8	16.3	30	12.0	4.0	49.0	17.6	14.4	12	0.759
NT-pro-BNP	403.0	35.0	28,963.0	1603.3	5106.2	32	538.5	12.0	3636.0	1040.3	1166.3	16	0.751
TP	64.5	46.0	78.0	63.8	8.1	44	68.5	59.0	84.0	68.0	5.7	28	0.034
Albumin	36.0	22.0	45.0	35.9	6.0	43	43.5	32.0	49.0	42.1	4.3	28	<0.0001
pH	7.4	7.3	7.6	7.4	0.0	33	7.4	7.3	7.5	7.4	0.0	21	0.118
pO_2_	7.8	5.3	11.1	7.8	1.5	33	8.6	6.8	14.9	8.8	1.6	21	0.018
pCO_2_	4.8	3.4	12.7	5.0	1.6	33	4.6	3.7	5.9	4.7	0.5	21	0.478

**Table 6 life-12-00357-t006:** Laboratory parameters in the group with infective causes of admission (both the ongoing- and post-COVID groups).

	Infection												*p*
	**On Going**						**Post COVID**						
	**Median**	**Min**	**Max**	**Mean**	**SD**	**N**	**Median**	**Min**	**Max**	**Mean**	**SD**	**N**	
CRP	92.5	3.0	333.0	109.3	86.9	38	66.0	6.0	285.0	92.3	71.9	19	0.531
WBC	10.1	3.4	75.0	13.5	12.0	38	11.3	4.2	19.8	11.6	3.7	19	0.588
PCT	0.2	0.1	49.0	6.0	15.3	10	0.6	0.2	1.0	0.6	0.6	2	1.000
D-dimer	1700.0	366.0	10,375.0	2776.7	2706.2	25	900.5	248.0	4471.0	1101.8	1114.1	12	0.003
Urea	5.9	2.4	23.9	7.6	5.0	38	5.3	3.6	15.5	7.1	3.6	19	0.872
Creatinine	89.0	49.0	239.0	96.8	44.3	38	73.0	6.1	125.0	74.3	26.3	19	0.073
Troponin	17.5	4.0	71.0	21.5	17.6	20	8.0	3.0	45.0	13.5	12.6	10	0.094
NT-pro-BNP	237.0	35.0	28,963.0	1813.0	6401.0	20	748.5	44.0	4617.0	1253.7	1578.8	12	0.039
TP	65.0	46.0	78.0	64.1	9.0	30	63.5	51.0	72.0	63.2	5.9	14	0.677
Albumin	35.0	22.0	42.0	34.3	5.6	29	42.0	27.0	45.0	39.1	5.8	14	0.003
pH	7.4	7.4	7.6	7.4	0.0	23	7.5	7.3	7.5	7.4	0.1	10	0.398
pO_2_	8.0	5.3	11.1	7.8	1.4	23	7.0	5.8	11.1	7.8	1.7	10	0.570
pCO_2_	4.8	3.5	6.6	4.8	0.8	23	4.8	3.4	12.7	5.6	2.6	10	0.583

**Table 7 life-12-00357-t007:** Non-pulmonary complications in the groups of patients.

Complication	Entire Sample	Ongoing-COVID Group	Post-COVID Group
Cardiac	11 (11.2%)	6 (10.7%)	5 (11.9%)
Arrhythmia	6 (6.1%)	3 (5.4%)	3 (7.1%)
Cardiac failure	11 (11.2%)	6 (10.7%)	5 (11.9%)
Renal failure	11 (11.2%)	8 (14.3%)	3 (7.1%)
Myopathy	2 (2.0%)	2 (3.6%)	0
*Clostridioides difficile* infection	6 (6.1%)	5 (8.9%)	1 (2.4%)

**Table 8 life-12-00357-t008:** Non-pulmonary complications by main diagnoses of pulmonary complications.

Complication	Cardiac Total	Arrhythmia	Cardiac Failure	Renal Failure	Myopathy	Colitis (*C. difficile*)
Infection (total)	9	5	9	8	2	6
Ongoing-COVID	4	2	4	6	2	5
Post-COVID	5	3	5	2	0	1
COPD exacerbation, non-infective (total)	1	1	1	2	0	0
Ongoing-COVID	1	1	1	2	0	0
Post-COVID	0	0	0	0	0	0
Post-COVID-19 ILD (total)	1	0	1	1	0	0
Ongoing-COVID	1	0	1	0	0	0
Post-COVID	0	0	0	1	0	0
Pulmonary embolism (total)	0	0	0	0	0	0
Ongoing-COVID	0	0	0	0	0	0
Post-COVID	0	0	0	0	0	0
Sarcoidosis (total)	0	0	0	0	0	0
Ongoing-COVID	0	0	0	0	0	0
Post-COVID	0	0	0	0	0	0

**Table 9 life-12-00357-t009:** Comorbidities in the groups of patients.

Comorbidity	Entire Sample	Ongoing-COVID Group	Post-COVID Group
Hypertension	56 (57.1%)	35 (62.5%)	21 (50.0%)
Cardiac disease	30 (30.6%)	17 (30.4%)	13 (30.9%)
Respiratory disease (COPD, asthma, ILD, cystic fibrosis)	41 (41.8%)	24 (42.9%)	17 (40.5%)
Diabetes mellitus	38 (38.8%)	25 (44.6%)	13 (30.9%)
Nephropathy	17 (17.3%)	13 (23.2%)	4 (9.5%)
Autoimmune disease	8 (8.2%)	5 (8.9%)	3 (7.1%)
Malignancy	22 (22.4%)	14 (25.0%)	8 (19.0%)
Transplantation	5 (5.1%)	4 (7.1%)	1 (2.4%)

**Table 10 life-12-00357-t010:** Comorbidities by main diagnoses of pulmonary complications.

Comorbidity	Hypertension	Cardiac	Respiratory	Diabetes Mellitus	Renal	Autoimmune	Malignancy	Transplantation
Infection (total)	34	14	26	19	8	7	16	2
Ongoing-COVID	23	9	17	16	7	4	11	2
Post-COVID	11	5	9	3	1	3	5	0
COPD exacerbation, non-infective (total)	4	4	5	4	2	0	1	1
Ongoing-COVID	2	2	3	2	2	0	1	1
Post-COVID	2	2	2	2	0	0	0	0
Post-COVID-19 ILD (total)	13	8	7	12	5	1	3	2
Ongoing-COVID	7	4	1	5	2	1	1	1
Post-COVID	6	4	6	7	3	0	2	1
Pulmonary embolism (total)	3	3	3	3	2	0	2	0
Ongoing-COVID	3	2	3	2	2	0	1	0
Post-COVID	0	1	0	1	0	0	1	0
Sarcoidosis (total)	2	1	0	0	0	0	0	0
Ongoing-COVID	0	0	0	0	0	0	0	0
Post-COVID	2	1	0	0	0	0	0	0

## Data Availability

All data presented in this study are included in this article.

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
