# Peer review of "Pulmonary Complications after COVID-19"

_life, 2022, doi:10.3390/life12030357_

Round 1

Reviewer 1 Report

The paper focuses on pulmonary complications resulting from COVID-19 infection, a topic that is surely actual and of scientific resonance.
However, in my opinion, the manuscript requires a thorough review before being deemed publishable.

  • Introduction: this paragraph usually serves to provide the reader with the background on which the study is based. In this specific case, however, it is too long; much of the introduction, in fact, should be moved to the Discussion, especially the extensive citation of current literature.
  • Matherial & Methods / Results: the purpose of the study is not clear, the authors limit themselves to a mere list of complications, subdividing them on the basis of temporal criteria (ongoing-COVID vs. post-COVID). There is no trace of any statistical analysis aimed at clarifying whether there are significant differences between the two groups. Furthermore, there is a lack of patient demographics (gender, age, etc.).
  • Discussion: in light of the revision of the statistical analysis, the discussion should be modified accordingly.

Unfortunately at the moment I cannot consider it worthy of publication.

Reviewer 2 Report

General comments:

The authors demonstrated the clinical features of several complications in ongoing-COVID and post-COVID. Their article is likely to help readers to learn the complications after COVID-19 infection. However, there may be several biases because the present study may be conducted retrospectively in the small number of patients at a single center. One of impressive results may be a relationship between sarcoidosis and COVID-19 infection. Another one of them may be findings of bacterial infective complications. Because the paragraphs of introduction and discussion are too long to read, the original message may become tenuous in the present study. Although the review for each section has been adequately addressed, several changes are required to update the manuscript.

Specific comments:

Major:

#1. As mentioned above, since the paragraphs of introduction and discussion are too long to read, the paragraphs should be shortened by removal of unnecessary sentences in the text.

#2. According to the authors’ results, sarcoidosis may be one of risks for COVID-19 infection in the present study. Since sarcoidosis patients are received steroid therapy, which increases susceptibility to infection, the authors should demonstrate the background of sarcoidosis patients at the admission in the text. The scadding stage of sarcoidosis should be also demonstrated in the text. On the other hand, because the authors speculated that COVID-19 infection may activate latent sarcoidosis in the discussion, the disease behavior of sarcoidosis among the patients should be demonstrated in the text.

#3. Results: The present study demonstrated several tables. Are there any differences in causes of admission or complications between ongoing-COVID and post-COVID?

Minor:

#1. Results: Median values of age and period from PCR tests to admission should be demonstrated in the text. Are there any differences in age or period from PCR tests to admission between ongoing-COVID and post-COVID?

Round 2

Reviewer 2 Report

General comments:

The authors have updated the text in the revised version of their manuscript. Thanks for their clear response for each review. They removed their speculation that COVID-19 infection may activate latent sarcoidosis in the revised text. Almost concerns have been adequately addressed in the manuscript. The reviewer will be without any regret after the editor’s decision.